# A State Validation System for Augmented Reality Based Maintenance Procedures

**Federico Manuri** †, **Alessandro Pizzigalli** † and **Andrea Sanna** *,†

Dipartimento di Automatica e Informatica, Politecnico di Torino, corso Duca degli Abruzzi 24,
10129 Torino, Italy; federico.manuri@polito.it (F.M.); alepizzigalli@gmail.com (A.P.)

**\*** Correspondence: andrea.sanna@polito.it

† These authors contributed equally to this work.

**Abstract:** Maintenance has been one of the most important domains for augmented reality (AR) since its inception. AR applications enable technicians to receive visual and audio computer-generated aids while performing different activities, such as assembling, repairing, or maintenance procedures. These procedures are usually organized as a sequence of steps, each one involving an elementary action to be performed by the user. However, since it is not possible to automatically validate the users actions, they might incorrectly execute or miss some steps. Thus, a relevant open problem is to provide users with some sort of automated verification tool. This paper presents a system, used to support maintenance procedures through AR, which tries to address the validation problem. The novel technology consists of a computer vision algorithm able to evaluate, at each step of a maintenance procedure, if the user correctly completed the assigned task or not. The validation occurs by comparing an image of the final status of the machinery, after the user has performed the task, and a virtual 3D representation of the expected final status. Moreover, in order to avoid false positives, the system can identify both motions in the scene and changes in the camera's zoom and/or position, thus enhancing the robustness of the validation phase. Tests demonstrate that the proposed system can effectively help the user in detecting and avoiding errors during the maintenance process.

**Keywords:** augmented reality; maintenance procedures; state validation; computer vision

## 1. Introduction

The term augmented reality (AR) comprises all the software and hardware technologies that confer to the ability to experience both the real world and computer's generated contents at the same time. Since artificial and physical objects are mixed together, the user can move in a hybrid space without constraints. AR is part of the mixed reality category, as depicted in the reality-virtuality continuum defined by Milgram and Kishino [1]. An indisputable advantage that can be attributed to AR compared to virtual reality (VR) is that the user can keep contact with the real world. This advantage provides two kinds of benefits: Firstly, since part of the environment that the user sees is real, it is not necessary to compute a virtual model of it; secondly, since the user's physical point of view is preserved, the physical and mental annoyances that usually occur in a detached full-immersive virtual world are avoided. Overall, with its capability of bridging the gap between real and virtual worlds, AR can be considered the best solution every time it is necessary to represent real and computer-generated elements within the same space.

The concept of maintenance relates to the adequate care and actions required to ensure the correct operation of a given equipment. As a consequence of the technological improvement of such equipment, technicians involved in complex maintenance and repair tasks often need to refer to instruction manuals to correctly complete assigned procedures. It is widely recognized that the action

of continuously switching attention between two sources, such as the device under maintenance and the manual, might cause a high cognitive load [2]: This can lead to a higher occurrence of mistakes, thus repair times (and costs) can increase.

AR can efficiently tackle this issue as digital contents are overlaid and correctly aligned with respect to the device to be maintained and can be conveyed to technicians while they are performing the procedure. Benefits of AR support in maintenance, repair and assembly tasks have been deeply analyzed in [3] and other researches such as [4], which proved the cost reduction and performance increase that can be achieved through the adoption of AR.

Maintenance has been one of the most important domains for AR applications since its dawn [5]: AR applications enable industry technicians to receive visual and audio computer-generated aids while performing different kinds of activities, such as assembling, repairing, or maintenance procedures (Figure 1). Early examples of supporting technicians with AR-based systems, which can be dated back to the 1990's, were all based on special purpose hardware (e.g., Head Mounted Displays) [6,7]. A set of possible high-impact applications for industrial AR were identified by Navab [8] in 2004. Moreover, the benefits that AR could provide to maintenance, repair and assembly tasks have been thoroughly analyzed by Henderson and Feiner in [3].

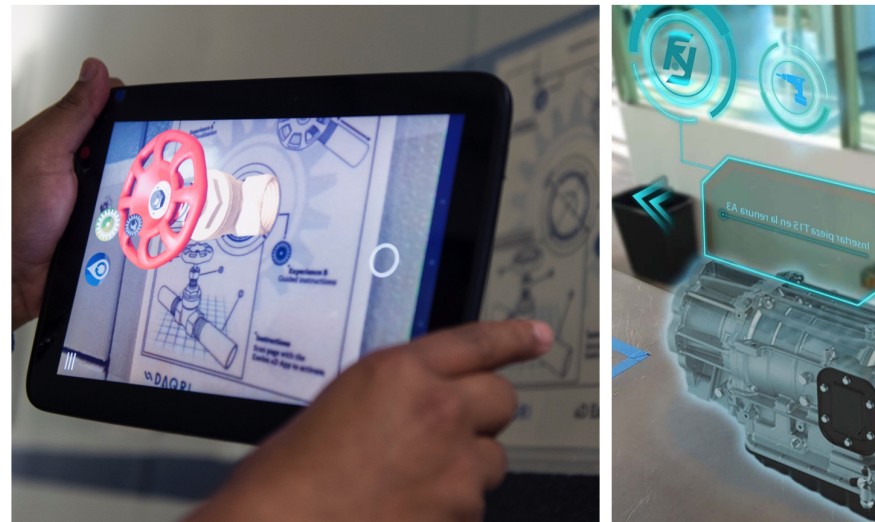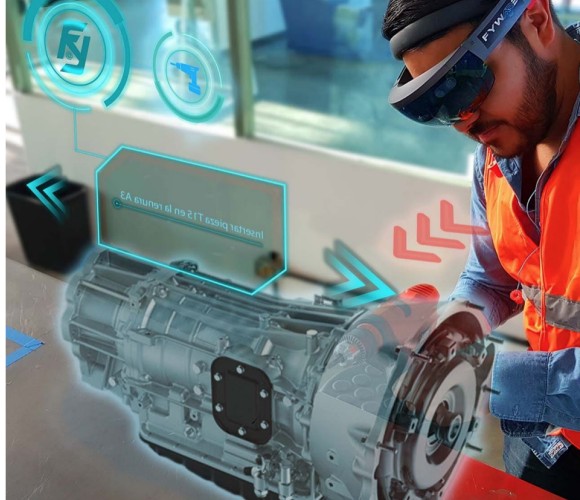

**Figure 1.** Examples of augmented reality (AR) applications for maintenance. **Left image** from Office of Naval Research, https://www.flickr.com/photos/usnavyresearch/23388133082, **right image** from Carlos Fy, https://commons.wikimedia.org/wiki/File:Entrenamiento-industrial-Fyware.jpg.

Currently, maintenance, repair and assembly are identified as strategic application fields, since the reduction of associated costs represents a key goal in many domains, thus, any technological advancement is carefully considered in order to take the opportunity to reduce these costs. Moreover, augmented reality is considered one of the nine pillars of technological advancement comprised by the Industry 4.0 paradigm: to provide an exemplification of the impact of such technologies on Industry, it has been evaluated that Industry 4.0 in the next 5–10 years will boost productivity among all German manufacturing sectors by 90 billion euros to 150 billion euros, possibly driving an additional revenue growth of about 30 billion euros a year [9].

A typical maintenance task can benefit from AR since it is possible to provide assistance to the user through digital assets: 3D models, animations, audio tracks and video clips, which are available to the user to better understand how to perform the given task. Moreover, AR applications for maintenance and repair are often completed by tele-presence systems: The user may request assistance to a remote expert that can interactively support the maintainer when AR aids are not sufficient. Usually, one of the main limitations of this kind of system is that users manually validate each step of the procedure.

Thus, they might wrongly execute or skip some steps, resulting in a significant waste of time which would greatly affect the cost of these procedures.

To the best of the authors' knowledge, the problem of automatically validating user actions has not yet been addressed by other works in the domain of augmented maintenance. Thus, the novelty of the proposed solution is that the system itself can automatically verify if the user has correctly performed the required operations or not. Maintenance procedures commonly require the user to interact with the machinery adding components, removing components or performing other actions that actually change the visibility of a piece of the machinery from a specific point of view, such as displacements. Framing the machinery from a static point of view, it is possible to recognize the key features that uniquely identify the different parts of the machinery, which are involved in the maintenance procedure. As it is known in advance what the user should do at a given step, it is possible to create a virtual model of the machinery that represents how it should look if the user has correctly performed the task. Thus, analyzing the video stream provided by the camera, it is possible to evaluate each change that should occur to the machinery. Then, comparing the framed scene with the virtual model of the machinery (framed from a virtual position corresponding to the real camera location) it is possible to validate the operation performed by the user.

The paper is organized as follows: Section 2 briefly outlines the state of the art of AR applications for maintenance procedures. Section 3 presents a detailed description of the proposed system, with a focus on the computer vision algorithms adopted and the technical problems that arose in the process of the development. Section 4 introduces the use cases selected for the assessment phase and presents the tests performed to evaluate the effectiveness of the system, whereas Section 5 provides an evaluation of the experimental data collected through the tests. Finally, open problems and future works are discussed in Section 6.

## 2. State of the Art

Maintenance is one of the major applications for AR since its early adoption [10]. The first attempts of supporting technicians using AR tools can be dated back to the 1990's, as detailed in [6,7], and the majority of these were based on special purpose hardware (e.g., HMDs).

The maintenance domain poses some specific challenges to the adoption of AR technologies [11]: Tracking is one of the most important and has been deeply investigated through the years. Real objects' detection and tracking in an industrial environment is a challenging task based upon the progress of image processing algorithms. Whenever it is unavailable, marker-based solutions are adopted. Since the early experiments by Feiner et al. [12] on the KARMA project, with a head-worn AR prototype designed to support end-users in performing simple maintenance procedures on a laser printer, marker-based AR applications have been widely investigated. Other important projects based on maker solutions are the ARVIKA project [13], the STARMATE (SysTem using Augmented Reality for Maintenance, Assembly, Training and Education) project [14] and the ManuVAR [15] project. However, the use of markers is an impractical solution in industrial environments, thus the tracking of the target objects should rely on their natural characteristics, such as textures or edges, identified through image processing. In 2006, the ARTESAS project (Advanced Augmented Reality Technologies for Industrial Service Applications) continued the research activities started by the ARVIKA project, focusing on three themes: Markerless tracking systems in industrial contexts, interaction methodologies with AR devices and industrial AR applications [16].

Another challenge is provided by the industrial environment, since the real case scenario and target object could be far from the original representations of the procedure, maintainers may need help to correctly perform their task. The Etälä project [17] proposed tele-assistance and AR to establish a communication channel between local maintainers and remote experts. A similar approach was adopted in [18], where AR and VR were used to remotely support both trainees and technicians. In [19], a collaborative framework solution is proposed to train technicians in assembly tasks of complex systems such as aircraft engines. Another example is provided by the EASE-R$^3$ project [20],

which aimed to develop an integrated framework for an easy and cost-effective repair, renovation and reuse of machine tools within modern factories. As part of this project, an AR framework has been developed to provide: AR aids for maintenance procedures, remote assistance from an expert technician, and re-configurability of the maintenance procedures. However, industrial environments may prevent the deployment of such solutions, for example due to network limitations.

Overall, to further increase the benefits of AR systems to the maintenance domain, it should be combined with other tools, such as real-time monitoring, fault-diagnosis and prevention systems. These tools may enhance the capabilities of technicians and thus improve maintenance performances. Automatic validation of the user actions could be one of these tools and, since markerless solutions are to be preferred in industrial environments, it could rely on the same image processing algorithms used to track the objects in the scene.

The number of research projects on AR for maintenance clearly implies that this is a challenging area of investigation. However, the design and development of a system that could validate the tasks performed by the maintainers, independently of the environment, is still an open problem. This paper proposes a system that overcomes this problem using a combination of computer vision techniques. Since the aim of the proposed solution is to validate the operations performed by the user at each step of the procedure, the system should provide three specific features: Identifying any change in the status of the machinery, comparing the current status of the machinery with a reference one, informing the user on the validation of each step.

## 3. The Proposed System

The basic assumption for the proposed system is that, usually, a maintenance procedure can be defined as a sequence of elementary operations, or steps. An 'elementary' operation is an action that cannot be described as a sequence of actions. Each step often corresponds to one among these three actions: adding an element to the machinery, removing an element from the machinery, displacing an element of the machinery. Figure 2 shows an example of maintenance procedure described as a state diagram of sequential steps. Each step of the procedure can be defined by two statuses, the first prior to the user action and the second one at the end of the action.

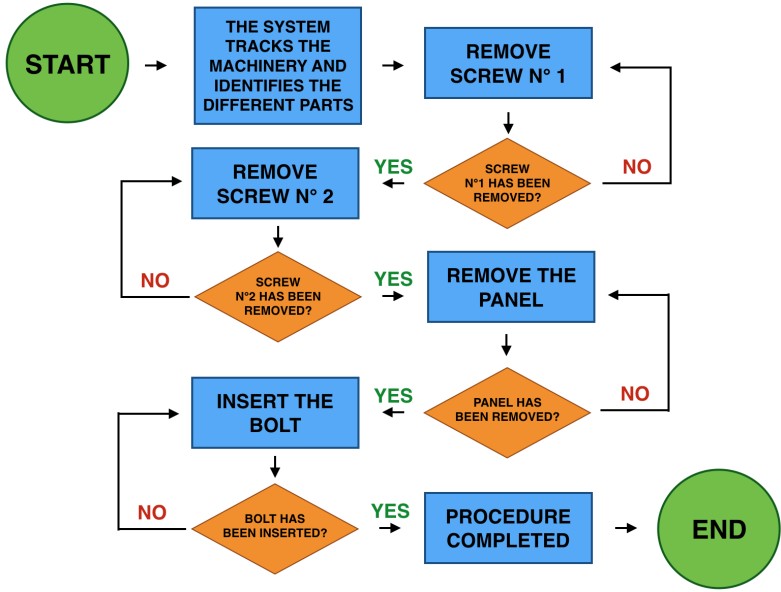

**Figure 2.** Example of state diagram describing the steps of a maintenance procedure.

In order to properly describe each step of the maintenance procedure, a peculiar set of rules has been adopted to illustrate the procedure through a textual file. Each line of the file defines one step of the procedure and provides four different parameters, delimited by tabulation characters: The step

number, the element to interact with, the final pose of the element (in case of a displacement) and the action required by the user. Moreover, since the aim of the proposed solution is to validate the operations performed by the user at each step of the procedure, the system should provide three specific features: Identifying any change in the status of the machinery; comparing the current status of the machinery with a reference one; and informing the user of the validation of each step.

*Software Architecture*

Since none of the existing AR Software Development Kit (SDK) provides all the functionalities required, the first step was to identify an open source computer vision library that could be extended in order to develop them. The ViSP library (Visual Servoing Platform library, https://visp.inria.fr/) has been chosen for the task, since it is an open source, modular cross platform library that provides visual tracking and visual servoing technics. Moreover, ViSP provides great flexibility both in terms of customization of the code and integration of other consolidated computer vision libraries such as OpenCV (Open Source Computer Vision Library, https://opencv.org/) or Ogre3D (Open Source Graphics Rendering Engine, http://www.ogre3d.org/). Figure 3 shows the software layers of the proposed system. The system has been deployed on a Windows desktop computer as a C++ application but, since all the involved libraries are available for OSX and Linux too, a porting would be possible. The video input stream is provided by a Logitech C905 Webcam (2MP image sensor, 3.7 mm focal length, f/2.0 lens iris, 640 × 480 resolution).

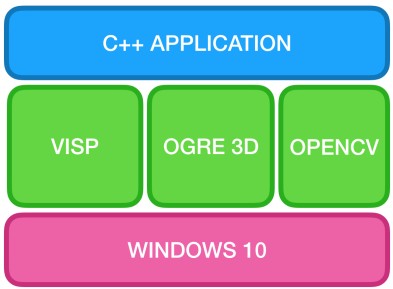

**Figure 3.** Software layers of the proposed system. ViSP (Visual Servoing Platform), OpenCV (Open Source Computer Vision Library), Ogre3D (Open Source Graphics Rendering Engine).

Tracking

The first step to deploy an AR maintenance system is to determine how to perform the tracking of the machinery and how to identify the elements that are subject to changes during the procedure. In order to avoid the limitations provided by the context where the machinery is deployed, a markerless model-based tracking system has been adopted. Among the three available markerless algorithms provided by ViSP, the vpMbEdgeKltTracker has been chosen, since it is a hybrid solution that provides a more accurate tracking of the object; vpMbEdgeKltTracker relies on both edge detection [21,22] and KLT (Kanade-Lucas-Tomasi) key points [23]. Edge detection algorithms usually perform well when objects without texture and with sharp and neat edges have to be tracked. On the other hand, the KLT (Kanade-Lucas-Tomasi) algorithm performs well when textured objects, with edges that could be difficult to recognize, have to be tracked.

In order to adopt a model-based tracking technique, it is necessary to obtain a Computer-Aided Design (CAD) model of the object to be tracked for each step of the procedure, which could be the whole machinery or a part of it. Moreover, 3D CAD models suitable for ViSP have to be saved as Cao files. Cao is a proprietary format developed for ViSP, which defines a 3D object as a textual description of lines, circles and cylinders that represent the object. The open source modelling tool Blender (https://www.blender.org/) has been used to model and to export CAD representations of the objects as Obj files. Since ViSP requires input geometries defined by Cao files, a Blender's Python script performing the conversion from Obj to Cao format has been implemented.

When the scene is framed by the camera, the system searches for the object to be tracked at the current step of the procedure, which could be the whole machinery or, if too big to be framed from a short distance, a part of it. However, all the three VISP's tracking algorithms consider all the feature key points for the framed object as a whole, thus, it is not possible to directly link a feature key point to a face of the CAD model. This limitation make it impossible to correlate feature points, provided by the algorithm, to a single element of the machinery that may change during the maintenance procedure. To overcome this issue, the original tracking algorithm has been modified in order to manage individually the different elements (the polygonal faces of the 3D CAD model) involved in the procedure. Since Cao files allow appointing each face of the object, it is possible to label the faces uniquely. Therefore, it is possible to define, for the chosen object, which are the elementary parts that are involved in the maintenance process, for each step of the procedure. The term 'component' is denoted as a part of the equipment under test, defined by a set of polygonal faces. If the maintenance operations have been correctly split into elementary steps, each step will usually affect only one component.

## 4. Workflow

The peculiarity of the proposed system consists of the sequence of image processing operations performed to validate the user's operations. Figure 4 shows the workflow of the overall system with respect to the steps of a standard maintenance procedure (green blocks): The automatic validation of the user's operation is repeated at the end of each step and only upon the user's request. The system consists of three image processing blocks (motion detection, status analysis and validation), an initialization block and a Graphical User Interface (GUI) to provide feedback to the user (blue blocks). The video input to the various blocks of the system is provided by a camera that frames the environment (in purple).

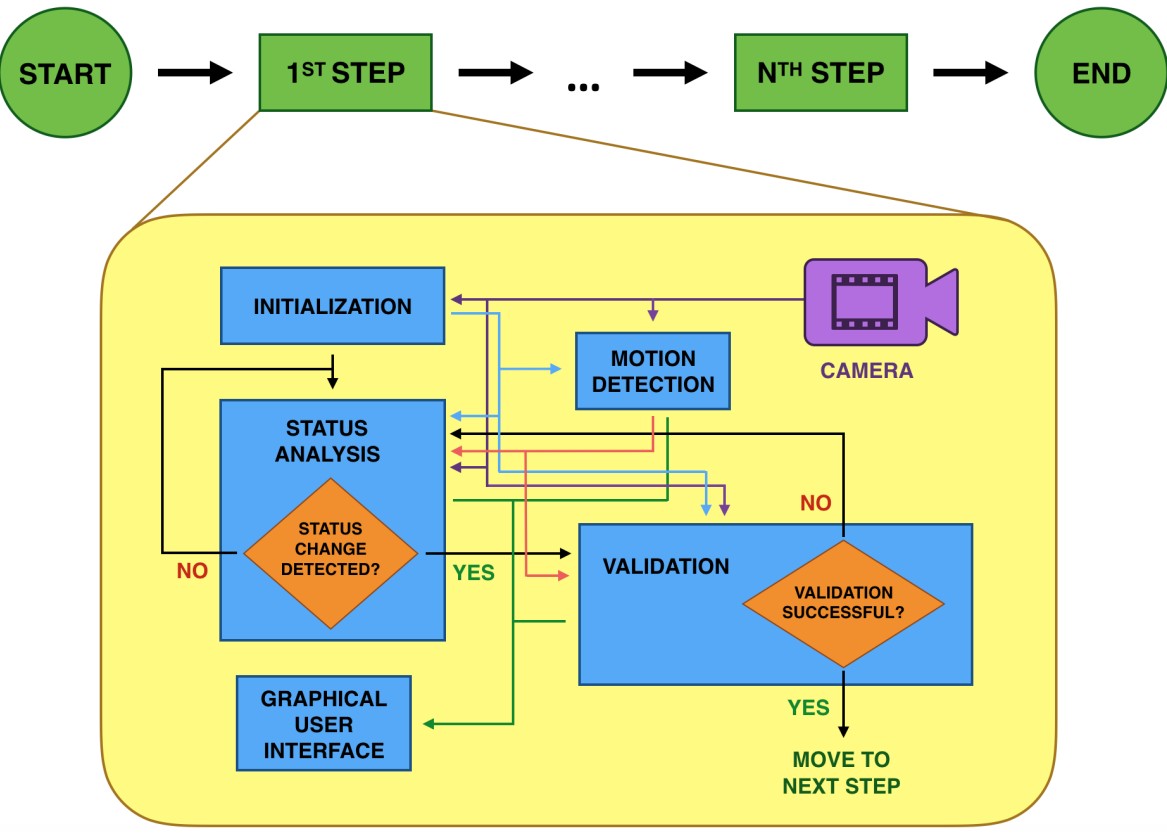

**Figure 4.** Workflow of the proposed validation algorithm.

A description of the overall system's workflow is provided here, whereas the details of each image processing block is explained in the following subsections. When the user request a step's validation, the initialization module performs some preliminary operations to enable the other modules to properly elaborate the frames from the camera. Then, the status analysis module performs the tracking of the machinery and evaluates the features of the tracked object in order to detect status changes' occurrences: If any change is detected, then the validation module is evoked to evaluate the correctness of the operation performed by the maintainer. Meanwhile, if any motion is detected in the scene, the status analysis and validation modules are instantly informed by the motion detection module (red arrows): This implies that any operation performed by these two modules will take into account this information in order to avoid errors. Validation will not be available till the motion stops and the data structures used for the status analysis and the validation phase will be cleared from any inconsistent value. Moreover, depending on the input and output of each module, the GUI module may be evoked by any of the previous modules to provide the user a feedback (green arrows). Finally, if a status change is detected and the validation is successful, the system can provide a positive response to the user and the maintenance procedure can proceed to the following step.

### 4.1. Initialization

The initialization phase takes place at the beginning of each validation and it is necessary to setup the tracking system. The most important role of this phase is to initialize the tracker, which is the virtual representation of the object to be tracked in the real world. The initialization module receives as an input:

- A Cao file representing the 3D model of the machinery that should be tracked at the current step;
- The video input provided by the camera;
- An initialization file named init; this file contains the most relevant points for the chosen 3D model; these points will be used for the initialization of the tracker;
- The camera's intrinsic and extrinsic parameters.

Even if the 3D model is the same for all the steps of the procedure, a different file has to be loaded at each initialization phase: This is necessary since all the faces which are not relevant for the current step are labeled as No_tracking, whereas the relevant faces are labeled by name. Thus, the labelled faces identify which component the user should interact with during the current maintenance step.

In order to correctly initialize the tracker, the user has to frame the target component with an angle and zoom that allow framing all the relevant key points defined in the init file. In order to help the user to perform this operation, the target object is shown in AR as a 3D wireframe model. When the target object is correctly aligned with the 3D reference model, the user can start the initializing procedure by pushing a button. At this point, the algorithm tries to initialize the matrix that defines the position of the model through the extrinsic parameters of the camera, enabling the system to match the 3D coordinates of the model with the 2D coordinates on the camera's frame.

The matrix has to be updated each frame in order to correctly evaluate the camera position with respect to the previous frame. If this operation is successful, the matrix coefficients and the initial pose of the tracker variable are correctly initialized; if the match between the virtual model and the real one is not satisfied, it is possible to repeat the initializing procedure. Otherwise, the matrix will be also stored for future use. This is useful if the camera and the target object have not been moved between two consecutive sessions of usage of the application, thus simplifying the initialization phase.

### 4.2. Status Analysis

To properly identify status changes in the tracked model it is necessary to recognize changes in a specific part of the model. At each step of the procedure, the technician will interact with only one element of the target object. Thus, the system will compute the element status prior to the technician interaction, then the final status will be computed when the technician completes the interaction with

the target object. In order to obtain two comparable statuses, the technician should guarantee that the position of both the camera and the target object does not change between the two status computation. Otherwise, the system would be unable to correctly compare them. The computation of the feature key points of the whole model could be challenging, depending on its size and on the complexity of the 3D model representing it. For this reason, instead of considering the whole target object, the status comparison only involves the key points pertaining to the position of the element that should change at a given step of the procedure. Thus, it is crucial to associate the feature key points provided by the tracking algorithm to the faces/polygons they are related to, in order to consider only those relevant for a given step. To obtain this result, the tracking algorithm provided by ViSP has been modified as follows: When the object is tracked by the algorithm, all the faces related to the element involved in the current step are projected in 2D, then the minimum and maximum x and y coordinates are computed to define the bounding box area for that element. As a result, it is possible to evaluate if a key point detected in the scene belongs to the element of interest or not. Only the key points whose coordinates are enclosed into the bounding box are taken into account at a given step of the procedure, whereas all the other key points are discarded. Thus, as the tracking algorithm analyzes the scene, all the key points that remain 'steady' are used to define the status of the specific component they belong to. A key point is considered 'steady' if it maintains the same position for a number of frame equal to the frame rate of the camera. Thus, if the camera frame rate is 24 fps, a key point has to be detected for at least 24 consecutive frames in the same position to be considered 'steady'. To evaluate the steadiness, the position of each key point detected at a given frame is compared with the coordinates of the key points detected at the previous frame. If there is a match, then the key point is considered steady and its frame counter is updated. When it reaches a value equal or higher than the frame rate, the key point is marked as steady. Two key points are considered equal if the position of the key point at a given frame is enclosed into a bounding box of 1 pixel with respect to the same key point at the previous frame. This helps to properly account for the errors provided by the camera quality and resolution and the imperceptible movements that may be involved. Figure 5 shows an example of key point detection (left image), positive comparison with a key point from the previous frame (center image) and negative comparison with a key point from the previous frame (right image). Table 1 shows the data structures used to store all the information relevant to identify and to analyze the status of a component. The left table contains the maximum and minimum values for the x and y coordinates of the bounding box defining the component. The right table lists all the feature key points that are identified inside the bounding box at a given time and it is updated at each frame. Each key point with a count value (which is the difference between the first frame and the last frame it was detected in) higher than the camera's frame rate (24 fps) is considered a steady feature (in the example, the first four key points from the top).

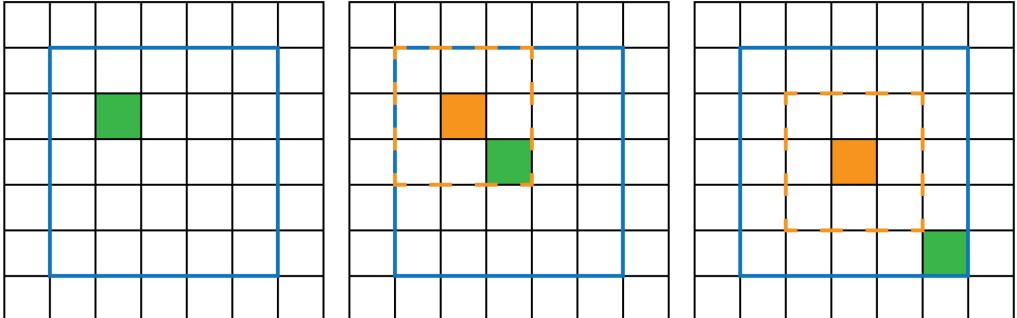

**Figure 5.** Key point's detection example: In the left image, a key point (green square) is detected for a given pixel, inside the bounding box that delimit the face of interest (blue box); in the center image, at the following frame, a key point is detected at one pixel distance respect to the previous one (the orange square), thus it is considered the same key point because it inside the 1-pixel bounding box (orange dashed box); in the right image, at the following frame, a key point is detected but it is outside of the 1-pixel bounding box of the previous frame, thus it is considered as a new key point.

**Table 1.** The bounding box's coordinates (left table) and the key points detected from the first frame to the 45th frame (right table). For each key point which is inside the bounding box's coordinates, the count value (which is the difference between the first frame and the last frame it was detected in) determine the steady features.

| BOUNDING BOX | |
| --- | --- |
| X_max | 7 |
| X_min | 1 |
| Y_max | 10 |
| Y_min | 1 |

| FEATURE KEY POINTS | | | | |
| --- | --- | --- | --- | --- |
| X | Y | LAST FRAME | FIRST FRAME | COUNT |
| 2 | 2 | 45 | 1 | 44 |
| 6 | 9 | 45 | 7 | 38 |
| 6 | 3 | 45 | 7 | 38 |
| 4 | 8 | 45 | 12 | 32 |
| 3 | 5 | 45 | 22 | 23 |
| 2 | 7 | 45 | 25 | 20 |
| 1 | 9 | 45 | 45 | 1 |

Thus, a component status is defined by the position and number of key points identified for its bounding box. When the technician completes the current operation and the system recognizes the view as steady (no motion), then the key features points are computed for the final status. This operation is usually quick, whereas instead most of the computational time is dedicated to the features comparison between the initial and final status. Thus, each key feature is compared based on its position (x,y coordinates), and if less then a threshold value of the correspondent key features is detected, then a status change is detected, since a significant number of key points have changed between the two statuses.

The threshold value has been obtained experimentally through performing numerous tests with the system, and it defines the minimum number (in percentage) of key points that should disappear from (or appear into) a face's bounding box to cause a status change, which has been computed as 30% of the key points for a given face. Overall, all the information pertaining the key points are stored in a specific structure: x and y coordinates, frame count for evaluating the steadiness and the frame number of first detection. This structure is updated each frame, removing key points that are no more detected, updating the frame count for steady key points and eventually changing the coordinates, if one pixel's movement is detected between two consecutive frames.

*4.3. Validation*

The validation phase has the goal to verify if the status change identified by the system corresponds or not to a correct operation performed by the user. In order to verify if the operation was performed correctly, it is necessary to compare the real object after the status change with the expected final status for the current step of the procedure. In order to properly compare the real scene with the 3D representation of the final status, it is necessary to define some guidelines. If the environment is known in advance, it would be possible to create a close representation of the real scene in Blender. If this is not the case, it is necessary to assume that the real environment is properly lit, with a stable and uniform light which minimizes shadows, glints, flares or spotlights on the target object. This is necessary to perform a fair comparison between real scene and 3D scene. Assuming that it is possible

to interact with the illumination features of the working environment, it is necessary to avoid all the setups that would introduce noise in the scene. Figure 6 shows the workflow of the validation module.

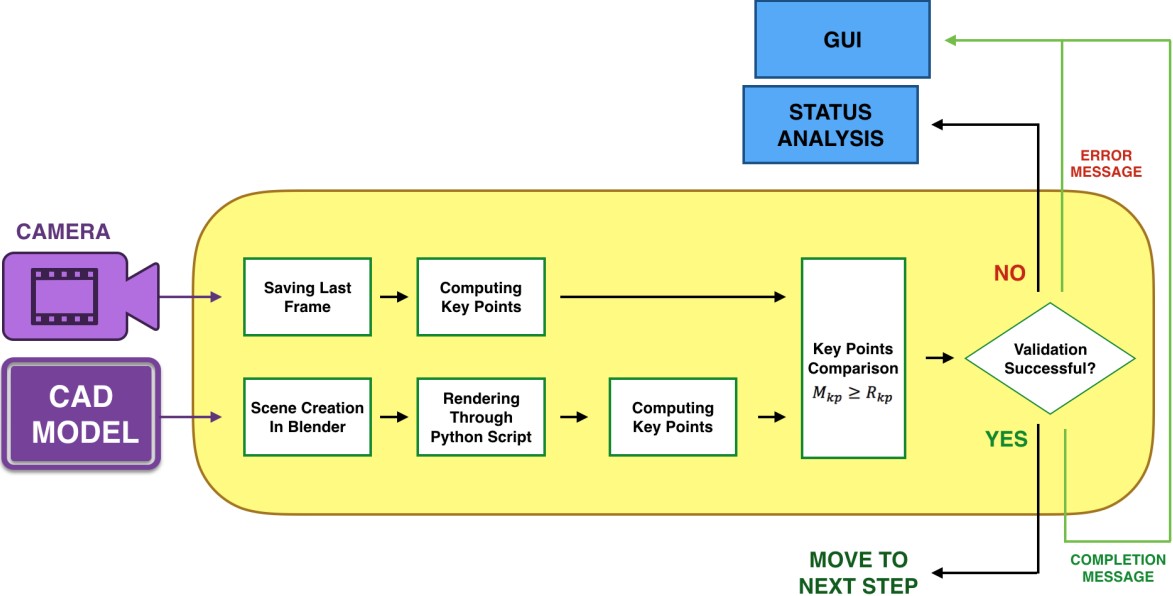

**Figure 6.** Operations performed by the validation module.

First of all, the validation algorithm takes a snapshot of the scene, right after a status change is detected. This snapshot will be used as reference for the real object final status. Then, a virtual representation of the expected final status is created taking into account the following variables:

- The camera's specifications;
- The camera's intrinsic and extrinsic parameters;
- The coordinate system provided by the tracking system;
- The object coordinates;
- A 3D model representing the final status of the object.

Since the proposed solution relies on a tracking system independent of the environment, it is not possible to have a prior formal description of the environment and of the lights surrounding the object. Lights have been added in Blender to the virtual scene in order to increase the scene's realism, providing shadows and environment illumination. Finally, the scene is rendered to obtain an image that could be compared with the snapshot of the real object; the rendering is performed in Blender through a Python script.

The snapshot and the render are then processed by the tracking algorithm in order to extract the two sets of feature key points, $S_{kp}$ and $R_{kp}$. Then, the number of key points that are equals between the two sets is computed ($M_{kp}$). Two key points are equals if they occupy the same 2D coordinates relative to the object's bounding box. Since the rendered image provides a lower level of detail compared to the real image, it is reasonable to assume that the tracking algorithm will find a lower number of key points for the rendered image. Moreover, even if a lot of parameters have been taken into account to render the scene as similar as possible to the real one, some differences from the two images should be expected. Since such differences may preclude a meaningful comparison, experiments have been carried out to identify a reasonable tolerance value that would minimize both false negative and false positive comparisons. Each key point in the render is compared with the key points at the same coordinates in the snapshot with a tolerance of 4 pixels. Thus, a pixel is considered matching even if the coordinates in the virtual image are displaced to a maximum of 4 pixels in any direction. Due to this tolerance, given a key point in the render, it could provide a positive comparison with more than one point in the snapshot, which will have a higher number of key points.

In order to correctly perform the validation phase, it is necessary to compare the number of matching key points ($M_{kp}$) and the number of key points detected on the render ($R_{kp}$). Since the virtual model will always provide a lower number of key points due to its level of detail, if $M_{kp}$ is equal or higher than $R_{kp}$, then the two images represent the same status, as defined by Equation (1):

$$M_{kp} \geq R_{kp} \tag{1}$$

If the validation result is positive, the user can proceed to the following step of the maintenance procedure; otherwise, the system will wait for another status change identification and then it will perform the validation phase again.

### 4.4. Motion Detection

When the user performs the procedure, false status changes could be recognized due to two possible reasons. Firstly, if a motion happens inside the framed scene, such as the movement of the user's hands. Secondly, if the distance and/or position between camera and tracking object changes, which may happen both if the camera or the object are moved or if the zoom factor of the camera is changed. In order to avoid these false positives in recognizing status changes, a mechanism of motion detection has been introduced. If at least one of the possible cause of motion is detected, then the validation system is halted till the motion end. Figure 7 shows all the operations performed by the motion detection modules.

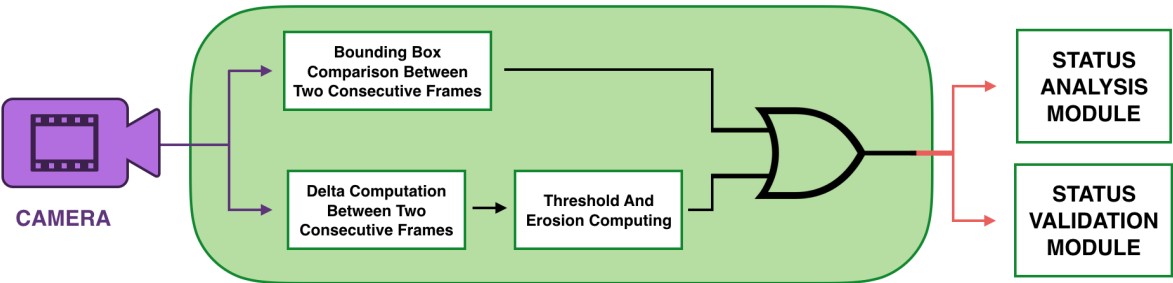

**Figure 7.** Motion detection schema.

It is possible to define the motion inside the scene as a transitory state between two static poses, since usually the workflow of a maintenance procedure requires the user to:

1. Frame the machinery, thus obtaining a static pose;
2. Learn through the assets which is the operation to be performed for the current step;
3. Interact with the machinery, thus generating a movement inside the scene.

To avoid false recognitions of status changes, motion detection inside the scene is performed through an algorithm that adopts some functions provided by OpenCV. The proposed method is based on performing the difference between two subsequent frames and applying a thresholding on the result. First of all, the two frames are converted to grayscale and the difference is computed with the OpenCV's absdi function. Then, the thresholding is applied to the result of the difference. The final image will be completely black if the two images are equals, whereas white dots will identify the differences between the two images, thus implying that a movement occurred.

In order to lower the number of false positive, such as the ones due to imperceptible movement of the camera, two different techniques have been adopted. Firstly, an erosion morphological operator has been applied to the result of the thresholding. Erosion allows deleting or reducing objects in a binary image, removing the smallest details. The resulting image is cleaner and presents less noise. Secondly, standard deviation has been computed for the image in order to obtain a value that actually describes motion distribution. Standard deviation will be equal or near zero if the motion affects only one pixel, such as the movement of a faraway object in the background. Otherwise, if most of the

framed area is affected by motion, then the standard deviation value will be high. The value of the standard deviation is then taken into account to decide if a motion in the scene has occurred.

The second issue was to identify camera motion, object motion or camera zooming. To this end, an algorithm has been developed to compare the coordinates of the bounding boxes that defines the distinct parts of the machinery at subsequent frame rates. When the tracking algorithm identifies the machinery, given its coordinates in the reference coordinate system, it is possible to compute the position of the bounding boxes which define its parts. If, at any next frame, one or more of these bounding boxes moves over at last one axis for more than a minimum threshold, as defined by Equation (2), then a movement is detected, as shown in Figure 8.

$$\sum_{c=x}^{y} \sum_{j=min}^{max} \frac{(c_{jt} - c_{j(t-1)})\backslash th}{(c_{min(t-1)} - c_{max(t-1)})\backslash th} = 0 \tag{2}$$

Each bounding box is defined by four values, $x_{max}$, $x_{min}$, $y_{max}$ and $y_{min}$. In the equation, $c$ represents the $x$ or $y$ coordinate of the bounding box at the current frame $t$ or at the previous frame $t - 1$. The variable $th$ is the chosen threshold whereas the $\backslash$ operator represents the integer division for $th$. The threshold value $th$ has been obtained experimentally performing numerous tests with the system and it defines the minimum movement along an axis that should be recognized to detect a motion. This value has been computed as 5% of the length of the bounding box side for a given axis.

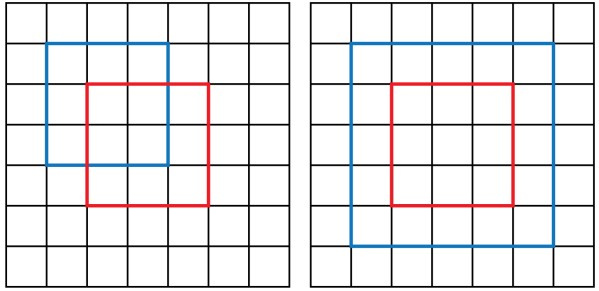

**Figure 8.** Examples of motion detection: the grid represents the pixels of the scene, the blue rectangle define the current bounding box for the reference object and the red rectagle define the position of the same bounding box at the previous instant. On the left, the system detect a motion due to either a pan of the target or a pan of the camera along both axis. On the right, the system detect a motion due to either a zoom out of the camera or the camera moving away from the target or the target moving away from the camera.

### 4.5. The User Interface

The first screen provided by the application allows the user to choose between examining a digital instruction manual or starting the maintenance procedure. The digital instruction manual has been inserted in order to allow the user to check how to implement each step of the procedure. The manual offers a recap of the pieces involved in the procedure, sorted by name. A textual content specifies which action should be performed in the current step, with two images defining the initial and final state/virtual representation of the model to be tracked. Arrows allow the user to move through the available steps of the manuals.

When the user starts the maintenance procedure, after the initialization step, the User Interface (UI) shows the machinery as it is framed by the camera, with the over imposed profile of the CAD model over it (Figure 9 in blue) as it is recognized by the tracking system. At the top of the UI a sentence describes the task to complete for the given step. 3D models such as arrows are used to better explain to the user where he/she has to operate. Three buttons are available: The first one allows the user to open up the digital manual; the second one is used to reinitialize the system (when the 3D model and the real machinery are not correctly aligned); the third one allows the user to hide or show the over imposed assets, depending on the user's preferences.

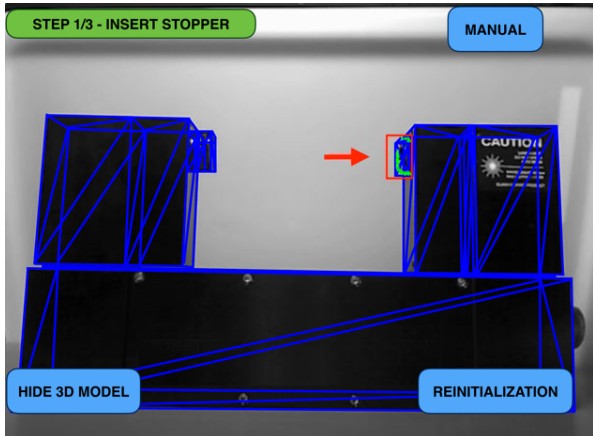

**Figure 9.** The graphic user interface.

When the system identifies a change in the object status, it is possible to check if the task has been performed correctly pushing the validation button. If the status of the object corresponds to the final status for the current step, the validation is successful and it is possible to proceed and an arrow button will appear, thus allowing the user to proceed to the following step. If the validation fails, it would be necessary to repeat the current task in order to fix the problem. If the currently completed step is the last one of the procedure, a notification of procedure completion appears.

## 5. Framework Evaluation

In order to deploy and evaluate the proposed solution, the blocks that compose the proposed system have been tested and verified; for this purpose, two use cases have been identified.

### 5.1. Use Cases

#### 5.1.1. Industrial Calibrator

The first use case is based on an actual industrial instrument, a laser calibrator. This tool measures, through a laser beam, the condition of molding tools in order to evaluate their precision during the lifespan and eventually suggests their replacement. Usually, these tools are used in industrial context where dust, chippings and other scraps from the machinery processing can fill or cover the laser lenses. In this situation, a specific procedure to clean the lenses of the calibrator is necessary to restore its working state. To evaluate the proposed system, a procedure to perform the lenses cleaning of the calibrator has been proposed. Figure 10 shows the 3D render of the industrial laser calibrator model: the red blocks identify the parts which should be modified by the user during the procedure. The procedure consists of the following four steps, that should be performed in reverse order to reassemble the calibrator after performing the cleaning:

1. Removing the cap;
2. Unscrewing the four screws;
3. Removing the external cover;
4. Pulling down the shutter to expose the lenses for cleaning.

Unfortunately, only 3 steps out of 4 could be properly validated by the system; due to the small dimension of the screws, it was not possible to retrieve enough key features to track them effectively.

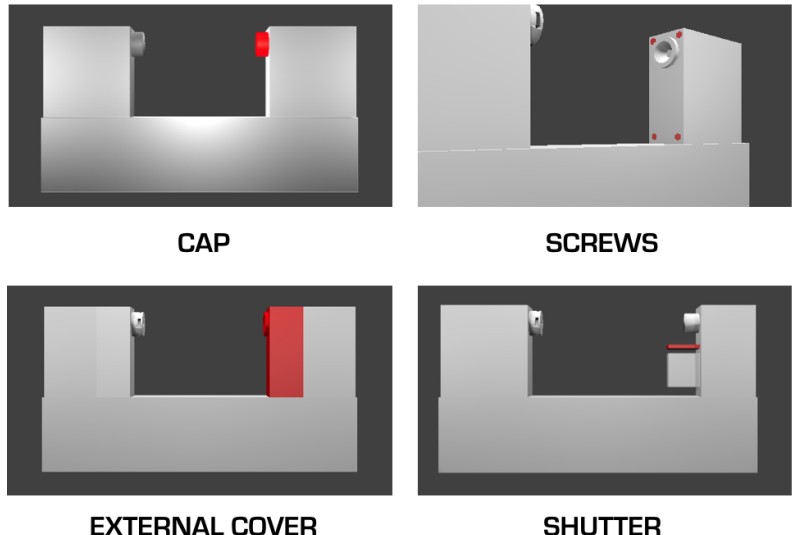

**Figure 10.** Relevant components for the industrial calibrator procedure.

### 5.1.2. Lego Model

As a result of the problem encountered in correctly tracking small objects such as screws, a second use case has been introduced to better evaluate the precision of the systems. This use case is based on a Lego model with mechanical components. A Lego model allows the use of pieces of variable dimensions, thus evaluating the precision when the system has to cope with small pieces. Moreover, Lego pieces allow performance of all the elementary operations accounted by maintenance procedures. The Lego model is made of a simple base with a set of blocks placed on top of it. Figure 11 shows the 3D render of the Lego model. The red blocks identify the parts which should be modified by the user during the procedure. Each of these blocks has been labeled uniquely; this allows, for each step of the procedure, to define which elements are relevant. The proposed use case is made of three steps:

1. Removing element Plpd2x1_start;
2. Moving the Trapdoor element, from position Trapdoor_close to position Trapdoor_open;
3. Adding element Plpd2x1_end to the Lego model.

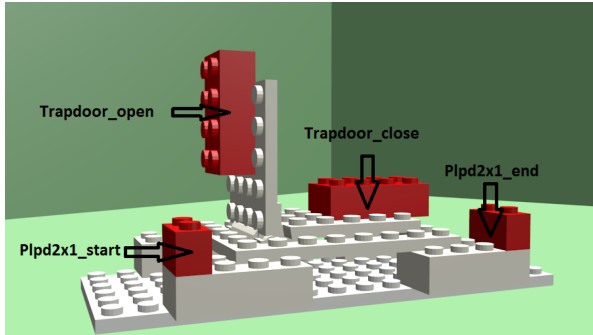

**Figure 11.** Relevant components for the lego procedure.

### 5.2. Tests

In order to correctly evaluate the proposed solution, each of the block that compose the proposed system has been tested and verified by the proposed user cases.

### 5.2.1. Feature Computation

The feature computation is the most important part of the algorithm. If features are computed incorrectly, the whole system will fail. Using ViSP, the computed features may be classified between

"detected correctly" and "detected with problems", which could be further classified depending on the problem, such as threshold, contrast, proximity, etc.

First of all, it is necessary to compute the bounding box of the target object. Then, the features detected correctly are saved. These features represent pixels that are different from their neighbors; usually pixels at the edges of objects. In order to obtain the best possible results, the machinery should be illuminated with a consistent light. The contrast between the machinery and the environment is important to avoid false positive features. Figure 12 shows how the object is correctly tracked before and after a displacement, thanks to the computation of the extrinsic matrix of the camera at each frame.

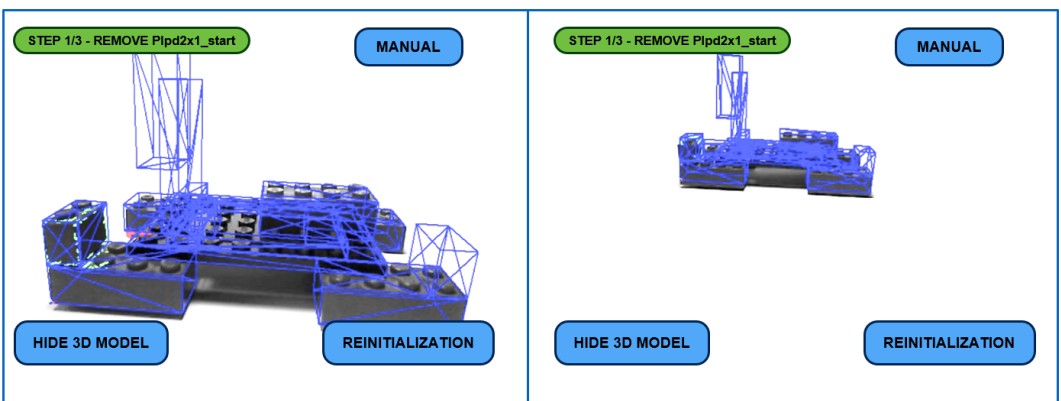

**Figure 12.** Object tracking when motion occours.

### 5.2.2. Object Insertion and Removal

Object insertion or removal consists in adding or removing a part to the machinery. In this case, the system validation is a crucial step in order to avoid wrong parts being added to the machinery, misplacement of parts or incorrect assembly. When the validation phase takes place, the status of the object is compared with the expected one. First, the features on real object are highlighted in green; then, all the common features computed by the system are highlighted in red. Thus, if there are enough red features compared to the number of green ones, it means that the two statuses are comparable and the validation will be positive. Figure 13 shows an example of validation phase: There are very few common features (highlighted in red), thus a failed validation occurs due to misplacement. On the other hand, Figure 14 shows a successful validation test. It is important to notice that even if the status of the object is the same as the expected one, not all the features are red. This is due to the fact that the total number of red key points on the left is higher than the total number of key points detected on the right image, even if it is not visible to the naked eye due to proximity or overlapping of such key points.

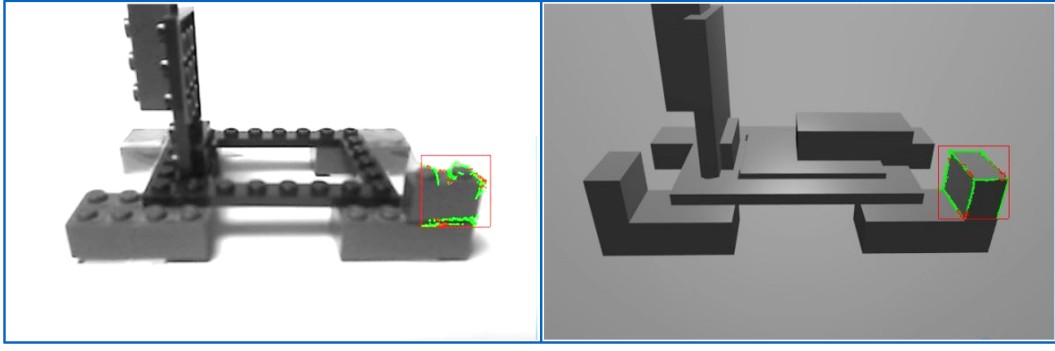

**Figure 13.** Unsuccessful validation phase: comparison between real object (on the left) and virtual model (on the right).

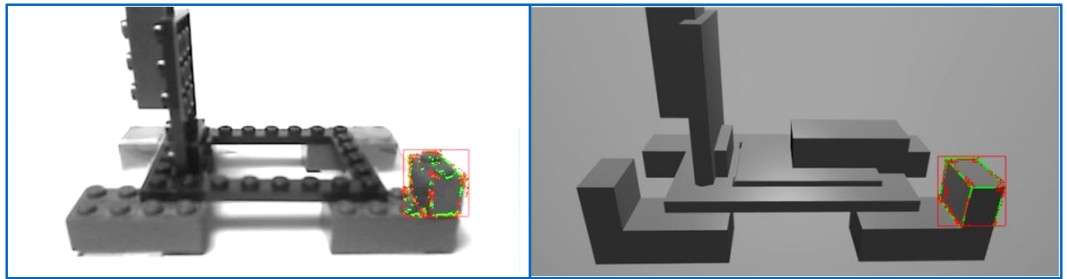

**Figure 14.** Successful validation phase: comparison between real object (on the left) and virtual model (on the right).

Figure 15 shows an example of a removal procedure with the industrial laser calibrator: First, the current object status is calculated (A), then the instruction for the given step are provided (B). After the external cover removal, the system computes the object status again (C). Since different key features are identified with respect to the initial status, the step is evaluated as completed (D), thus allowing the user to validate the current step or move to the next one.

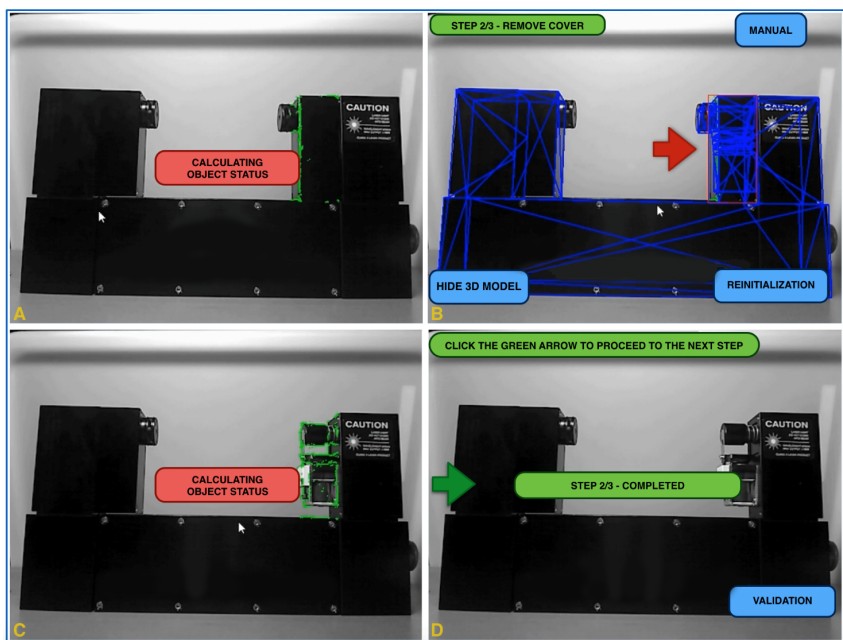

**Figure 15.** Industrial laser calibrator's external cover removal. The object status is calculated (**A**), the instruction are provided to the user (**B**), the system computes the object status again (**C**) and finally the step is evaluated as completed (**D**).

### 5.2.3. Object Displacement

The change of position of a piece on the machinery requires a more complex validation phase, since two conditions must be satisfied: First, the key points in the starting point of the piece should change, revealing that the piece has been removed; secondly, the key points in the final position should identify the new configuration on the machinery for the selected piece. Figure 16 shows how the feature computation takes place in the two different zones, with the black rectangle representing the bounding box referred to the starting position and the red rectangle the bounding box of the final one.

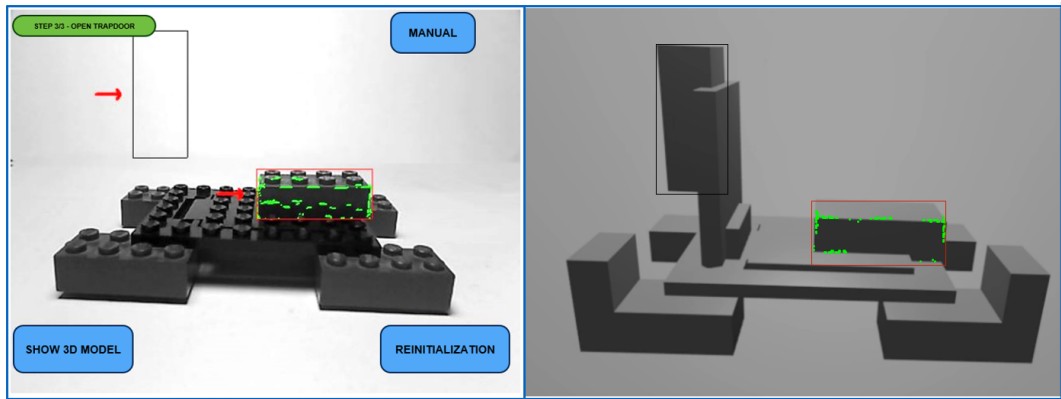

**Figure 16.** Starting position (red bounding box) and final position (black bounding box) for the given step, on the real object (**left**) and the virtual model (**right**).

Figure 17 shows another example of object displacement with the industrial laser calibrator use case. The user must move up the shutter (A), and both the initial and final position are depicted to the user (B). When the user performs the given task (C), neither false positive nor false negative status are identified. Finally, the system evaluates the object status and different key features are highlighted (D). It is important to consider that, even if the vertical displacement is minimal (less than a centimeter), the system correctly identifies different key features (green dots), as shown by Figure 17B,D.

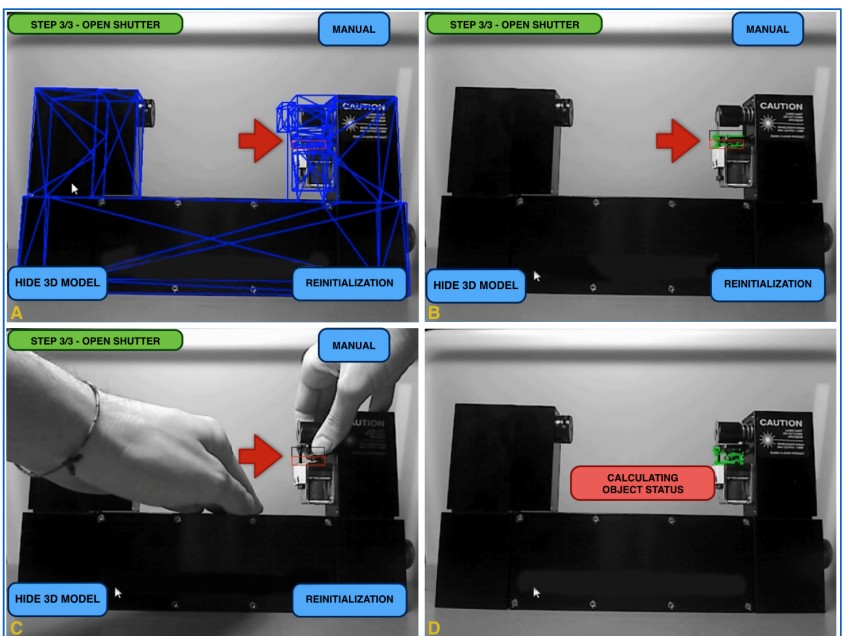

**Figure 17.** Industrial laser calibrator's shutters displacement. The object status is calculated (**A**), the instruction provided to the user (**B**), the user remove the external cover (**C**) and then the system computes the object status again (**C**).

## 5.3. Results Analysis

The tests performed with the proposed system allowed to evaluate both its precision and its limits. First of all, due to the number of features needed to correctly evaluate the status of the object, it may be difficult to correctly identify changes when working with small components. The smallest component involved in the test and correctly recognized by the system measured $8 \times 16 \times 11.2$ mm, corresponding to a bounding box area of $85 \times 90$ pixels (2,49%) for a $640 \times 480$ pixels frame of the scene. Thus, enhancing the proposed system with a High Definition or 4K camera will further improves the recognition precision of small objects in the scene.

Moreover, depending on the environment and the camera perspective, the correct computation of the features may be critical when the object removal operation is performed, because the algorithm might identify it as still present, thus missing a status change identification. The system may detect key features from the environment in the same area where the removed component was and perceive the scene as unaltered, thus leading to a false positive detection during the status analysis phase. The ViSP's tracking algorithm provides additional insights for feature identification, thus, it is possible to identify uncertain features and to classify them into two categories: Features with threshold problems and features with tracking problems. Comparing the features from these two groups it is possible to reduce the occurrences of false recognitions. Whenever the number of features that have problems is higher than the number of correct features, the component is recognized as missing, thus leading to a status change. However, in order to increase the system precision and to address the false recognition problem, it could be useful to use a depth camera to further improve the analysis of the scene's features.

Another relevant aspect is that the validation step is not real-time in terms of responsiveness. The time complexity is related to two operations, the status comparison and the rendering of the virtual scene. The first one is affected by the number of features to be compared for a given element, the second one is affected by the model complexity (e.g., number of polygons and textures) and the scene complexity (e.g., environmental lighting). Thus, the time complexity strictly depends on the specific use case due to the several variables involved. In the Lego use case, the 3D model has 182 polygons and a laptop computer with an AMD A6-3400M CPU and 6 GB of RAM has been used. In these conditions, the features computation requires 4–5 s whereas the comparison takes 10–11 s and the rendering 3–4 s. It was possible to make this phase optional, thus allowing the technician to perform the validation or to avoid it and proceed to the next step. However, a through improvement would be to provide real-time validation in order to always perform it without impacting on the performances and timing of the procedure.

Finally, in order to minimize the time needed by the different modules to perform the visual analysis of the scene, in each phase only the component involved in the current step is taken into account for the status analysis and validation steps. If the user accidentally performs an elementary operation on a different component than the one expected for the given step, depending on the area interested by the operation, the system may not be able to identify the status change, thus it would not be possible to provide the user any hint on what he/she is doing wrongly. To take into account all these conditions it would be necessary, at a given step, to always consider all the components that may be interested and/or modified by the user actions. This will not only improve the robustness of the system but may also improve the system usability, providing additional information when the user makes a mistake, such as informing the user when he/she works on the wrong component.

## 6. Conclusions and Future Works

In this paper, a system for supporting maintenance procedures through AR has been proposed. The novelty of the system consists in a computer vision algorithm able to evaluate, at each step of a maintenance procedure, if the user has correctly completed his/her task or not.

The validation occurs comparing the final status of the machinery, after the user performs the task, and a 3D representation of the expected final status. Moreover, the system is able to identify motion in the scene and to discriminate between motion in the scene or a change of position among the camera and the machinery.

The proposed system has been tested on two use cases, one involving an industrial laser calibrator and another one on a Lego model. The tests allowed to evaluate the precision and limitations of the proposed system: even if the system proves to be quite reliable, it could be enhanced to further improve its reliability and robustness. Future works will focus on further developing the recognition module to allow the correct tracking of tiny components. Another interesting development would be to take into account all the component of the machinery at each step of the procedure to further improve the robustness of the system.



**Author Contributions:** A.S. and F.M. wrote the paper and analysed data resulting from tests. A.P. and A.S. designed the algorithm, whereas A.P. wrote the code and performed the experimental tests.

**Funding:** This research received no external funding.

**Conflicts of Interest:** The authors declare no conflict of interest.

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
