# Peer review of "A State Validation System for Augmented Reality Based Maintenance Procedures"

_applsci, doi:10.3390/app9102115_

Round 1
Reviewer 1 Report
The paper address an important subject for AR application in an industrial context.
In my opinion sections 1. and 2. addresses extensively the state of art of AR, at the beginning the reader may think that it will be a paper reviewing the progress of AR in industrial scenarios.
The last paragraph from section 2. is confusing (lines 192-195), it can not be understood the meaning of such numbers in the text.
Mainly section 4. sounds like a technical report, it is suggested to reformulate it.
Lines 541-454 improving camera resolution is expected to "improve the recognition precision", what about framerate? If it takes too long to process an image the user will suffer from a high latency process, it should be considered too. Not only the image will be bigger but also the 3D model will become "heavier" to process.
Lines 556-557, be careful about depth cameras there are benefits but you may limit the distance from the camera to the target object to a shorter range of allowed distances.
About references, there are many of them not coming from scientific sources that did not provide a major contribution to the paper. It is suggested to review the state of the art section and its references.
It is also recommended to include a reference to “Augmented reality and the future of maintenance” (MPMM 2014 DOI:http://dx.doi.org/10.14195/978-972-8954-42-0_12) where the progress of AR in an industrial context is already presented with a useful timeline to easily understood the limitation of technologies and project for a given period. This will let the authors focus the state of art on significant methodologies that may support their research instead of giving an extended review of AR.
reference [45] is not a scientific one neither gives a significant contribution to the paper.
reference [46] is not available, without an alternative method to check it I suggest to remove it from the paper.
Author Response
Please find the author's note to the reviewer in the word file

Reviewer 2 Report
This paper presents a system that automatically validates AR-based maintenance procedures. The motivation and main issues are quite clear and properly explained with related works.
On the other hand, the technical novel contribution is not strong enough, and the proposed approach is quite heuristic and limited:
The proposed method is mostly based on pixel-wise image processing, which is very sensitive to noise so that it might not be appropriate for validation where its robustness and accuracy are critical and guaranteed.
- The status analysis is based on the bounding box with a range (1 pixel) and counting frames over time (actually, the KLT algorithm tracks feature points between frames, and their optical flow vectors can easily computed from them, without the proposed heuristic method).
- For the validation, comparing the number of feature points between the image and the rendered one is not convincing and very limited. The pixel-level tolerance (4 pixels) is not robust, as well. Finally, I am not sure how reliable the rendering scene is always as similar as possible to the real one.
- The motion detection is based on the frame difference, morphological operation, and standard deviation. Please explain how to distinguish between motions by the target object and motions by unwanted one (like hands) using the frame difference when they simultaneously happen.
- In the motion detection, there are ambiguities 1) when the object is moved while the camera is fixed or 2) when the object and the camera are moved at the same time.
In addition,
- The proposed method (such as analyzing status, validation) highly relies on feature points, but there are lots of less-textured objects, including ones in this paper and especially, industrial objects. As mentioned in this paper, the size of the object is problem, but its textures might be more significant.
- The face indexing might be challenging for complicated models, causing the failure of the object recognition as well as the high computational cost.
- The occlusions are inevitable during assembly procedures. Please explain how to handle them?
- Different from the experiments in this paper, the background is not ideally homogeneous, but normally cluttered.
- In section 5.3, the time complexity is mentioned, but should be evaluated in details (because the ViSP supports the real-time tracking, and it seems the proposed method is not based on simple processing, which requires the high computational cost). The supplementary video might be helpful, if possible.
- Experimental results are not sufficient to convince the proposed system.
Minors:
- Figure 4(left image): The vpMBEdgeTracker only uses edge information of objects, but why the features are extracted?
- Line 537: as shown by 22(B) and 22(D) --> 18(B) and 18(D)
Author Response

(The authors gave the same response as above.)

Round 2
Reviewer 1 Report
I accept the last revised revised version.
Reviewer 2 Report
The revision is acceptable, but it seems that the response keeps claiming that the proposed framework assumes well-controlled conditions, which might be far from in many general and more technical challenging cases.
In a practical manner, on the other hand, such well-controlled conditions might be possible on industrial cases, and also, this paper (and response as well) properly clarifies and discusses them.
Minors)
- As mentioned in A2, there is no explanation about the projection of the faces in this paper.
- As pointed out in Q10, it would be better to understand the time complexity when its details are shown in the paper.
Author Response

(The authors gave the same response as above.)

Round 3
Reviewer 2 Report
All responses are acceptable.